# The Paradox of Shorebird Diversity and Abundance in the West Coast and East Coast of India: A Comparative Analysis

A. P. Rashiba [1,†], K. Jishnu [2], H. Byju [3], C. T. Shifa [4], Jasmine Anand [5], K. Vichithra [6], Yanjie Xu [7], Aymen Nefla [8], Sabir Bin Muzaffar [9,*,†], K. M. Aarif [10,*,†] and K. A. Rubeena [11]

1 Department of Zoology, Wildlife Biology Division, Farook College PO, Kozhikode 673632, India
2 Department of Zoology, MES Ponnani College, Ponnani, Malappuram 679577, India
3 Centre for Advanced Study in Marine Biology, Annamalai University, Parangipettai 608502, India
4 Department of Zoology, Govt College, Madappally, Kozhikode 673102, India
5 Department of Zoology, T.K. Madhava Memorial College, Nangiarkulangara, Haripad, Alappuzha 690513, India
6 Department of Zoology, MES Keveeyam College, Valanchery, Malappuram 676552, India
7 The Helsinki Lab of Ornithology, Finnish Museum of Natural History, University of Helsinki, P.O. Box 17, FI-00014 Helsinki, Finland
8 Department of Biology, Faculty of Sciences of Tunis, University of Tunis El Manar, El Manar II, Tunis 2092, Tunisia
9 Department of Biology, United Arab Emirates University, Al Ain P.O. Box 15551, United Arab Emirates
10 Terrestrial Ecology, Centre for Environment and Marine Studies, King Fahd University of Petroleum and Minerals, Dhahran 31261, Saudi Arabia
11 Department of Biosciences, MES College Marampally, Aluva 683105, India
* Correspondence: s_muzaffar@uaeu.ac.ae (S.B.M.); achuarif@gmail.com (K.M.A.)
† These authors contributed equally to this work.

**Abstract:** Migratory shorebirds that move across continents along their flyways are undergoing a drastic decline globally. A greater proportion of them that regularly winter along the Indian coasts within the Central Asian Flyway (CAF) are also undergoing severe declines. However, the mechanisms underlying the population trends in these areas remain little understood. This study investigated the diversity, abundance, population dynamics and distribution patterns of shorebirds along the Indian coasts based on the available literature. The west coast of India is relatively less studied than the east coast in the CAF. Further, we observed that the diversity, abundance, population dynamics and distribution pattern of the shorebirds follow different trends on the west coast compared to the east coast. These variations are in accordance with the differences in topography and biotic and abiotic factors between the coasts. Anthropogenic activities have far-reaching effects on the survival and persistence of shorebirds along the coasts. The west coast is evidently more productive than the east coast at every trophic level and thus the west coast is expected to account for more abundance and diversity of shorebirds. Paradoxically, we found that the east coast supports a greater abundance and diversity of shorebirds than the west coast. The west coast, therefore, requires further investigations to obtain a better understanding of the causes of apparent differences in abundance and diversity as well as the observed declines in shorebirds, compared to the east coast of India.

**Keywords:** shorebirds; abundance; distribution; over-summering; east coast; west coast; conservation

## 1. Introduction

Shorebirds constitute a highly diverse group of migrant species that require a high amount of energy in association with their long-distance migration [1–5]. They are documented in nearly all shorelines of the world except Antarctica. The seasonal migration of shorebirds is an important biological event [6], characterized by long-distance travel among breeding, stopover and wintering sites driven by seasonal influences on resources [3,7–9].

Shorebird populations are declining worldwide. The drastic decline of shorebird populations in Eastern Canada and the north-eastern United States over the past few decades has been attributed to habitat degradation [10]. Out of the 25 shorebird species of the East Asian-Australasian Flyway (EAAF), 22 have undergone widespread decline and seven have been included in the IUCN Red List as Threatened or Near Threatened [11]. An overall decline in the population of Common Greenshank (*Tringa nebularia*), Temminck's Stint (*Calidris temminckii*), Wood Sandpiper (*T. glareola*) and Pacific Golden Plover (*Pluvialis fulva*) along the EAAF has been reported [12]. Similarly, significant declines in many shorebird populations along the Central Asian Flyway (CAF) have been linked to environmental factors, habitat loss or alteration influencing various trophic levels [13–15].

Although shorebirds that migrate along the Central Asian, South Asian Flyways, East Asian-Australian and Western Pacific Flyways use the Indian subcontinent as their primary wintering grounds [16,17], studies on shorebirds along coastal regions of India are very limited. India hosts major and key wintering grounds, stop-over sites, staging sites and over-summering sites for the migratory shorebirds along the Central Asian Flyway [14–17]. Further, they provide breeding grounds for Kentish Plovers, Black-Winged Stilts and Great Thick-knees. India, being a highly populated country, is significantly vulnerable to habitat degradation and loss due to anthropogenic activities and climatic changes.

Diversity patterns of waders including shorebirds have been studied in the Kole wetlands of Thrissur, a wintering site on the west coast of India, from 1998 to 2001 [18]. Spatio-temporal patterns of shorebirds at mangroves, mudflats and sandy beaches in Sindhudurg district, Maharashtra on the west coast [19] have also been studied. However, none of these studies have analyzed the population dynamics of shorebirds in depth. One study [14] documented the abundance of 15 migratory shorebird species over more than a decade at the Kadalundi-Vallikkunnu Community Reserve (KVCR), a coastal wetland in western India, which is a stopover and wintering area for many shorebirds species. This work highlighted the importance and the influence of relative humidity, air temperature, water temperature, salinity and invertebrate prey abundance on the departure dates of several shorebird species [14]. Furthermore, long-term changes in the nutrient content were found to be linked with prey abundance and induced declines in shorebirds in this region.

As the habitat characteristics are different between the east and west coast of India, the dominant shorebird species also vary [13,16,18–21]. For example, the dominant shorebird species found in Vedaranyam (Point Calimere), located along the east coast, were Lesser Sand Plover (*Charadrius mongolus*), Marsh Sandpiper (*T. stagnatilis*), Little Stint (*C. minuta*) and Curlew Sandpiper (*C. ferruginea*) [22] whereas those in the KVCR, located along the west coast, the dominant species were Plovers followed by Common Greenshanks and Common Redshanks (*T. totanus*) [13]. On the east coast of India, species composition, relative abundance and distribution patterns of shorebirds over two years from mudflats, tidal flats and freshwater habitats in the entire Pulicat Lake of Andhra Pradesh and Tamil Nadu [16] highlighted important threats encountered by shorebirds.

Species abundance of shorebirds on their wintering and stopover sites is influenced by local habitat characteristics, including vegetative structure, cover patterns, moisture and biomass associated with invertebrate prey [23–26]. Shorebirds use both natural and artificial habitats in winter as the shallow water at both coastal and inland wetlands provides a suitable habitat for them [27]. Tidal flats, especially mudflats, were recorded with higher density, diversity and richness of shorebirds in all seasons on the east coast [14,19,28]. Migratory shorebirds were attracted to prey species directly and to nutrients indirectly, which in turn was under the influence of rhythmic changes in tidal patterns. Higher shorebird density, diversity and species richness were documented during the migratory season than in other seasons on the east coast [14,19,28].

The uniqueness of distinct geographical regions, which includes food availability, nature and quality of the substrate, water quality, and other ecological features, determines the shorebird species composition, abundance and distribution [1,29]. Tremendous anthropogenic pressures like trapping, lime shell mining, pesticide contamination, al-

terations of coastal wetlands, short-term environmental changes, and long-term global climate change exert devastating pressures on these habitats and hence on the shorebird populations [30–32], particularly in Indian wintering grounds [16,17].

Studying the abundance and spatiotemporal patterns of shorebird populations is absolutely crucial in managing ecosystems [6] and hence is fundamentally important to conduct extensive studies to identify all the crucial wintering and stopover sites, seasons and habitats of shorebirds along the Indian coast in addition to the previously reported sites [19]. In India, the Arctic Migratory Birds Initiative (AMBI) along with the Conservation of Arctic Flora and Fauna, Ministry of Environment, Forest and Climate change, Bombay Natural History Society (BNHS) along with other relevant Indian institutions now facilitate and coordinate shorebird survey projects both nationally and globally. Every year, the shorebird counting program on world shorebird day (6 September) takes place between 2-6 September. It is an effort to raise awareness about the importance of regular shorebird monitoring and counting as the core element for the protection of bird populations and habitat conservation [11]. However, at present, studies on the population of shorebirds along the Indian coast have been carried out in only a few areas and a large part is yet to be explored.

The present study aims to carry out a comparative analysis, by a systematic literature review, of the diversity, abundance, population dynamics and distribution pattern of shorebirds on the west and east coast of India (Figure 1). The results aid in understanding the productivity of the coasts, which is functional in developing blueprints for conservation methods.

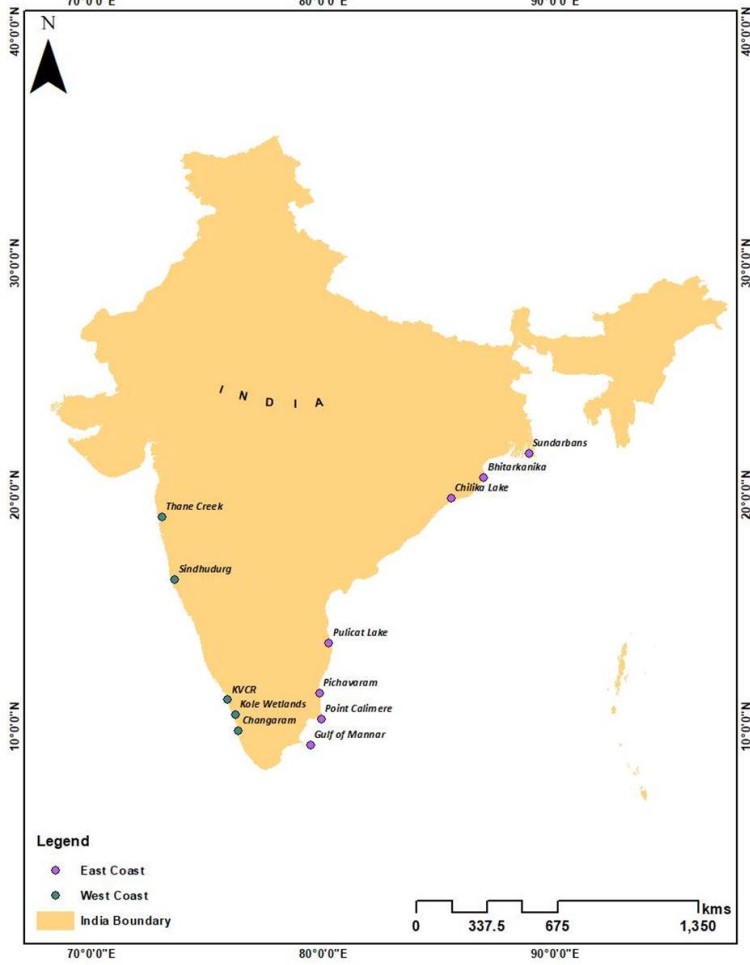

**Figure 1.** Map showing that major study conducted on shorebirds at west and east coasts of India.

## 2. Species Diversity, Abundance and Population Trends of Shorebirds on the West Coast

The west coast of India has been recognized as the important wintering ground for many shorebird species in the CAF, but the documentation on the diversity, abundance and spatiotemporal pattern of shorebirds from the west coast of India is scanty [13,17,19,33,34]. A total of 30 species of shorebirds belonging to the Charadriidae and Scolopacidae families were reported from Byet Dwarka Island, Gujarat [35]. Also, 12 species of shorebirds, including migrant, resident and resident migrant, have been recorded from Thane creek including the Black-tailedGodwit (*Limosa limosa*), Eurasian Curlew (*Numenius arquata*), Common Redshank, Marsh Sandpiper, Wood Sandpiper, Common Sandpiper (*Actitis hypoleucos*), Little Stint, Little Ringed Plover (*C. dubius*), Kentish Plover (*C. alexandrinus*), Lesser Sand Plover, Black-Winged Stilt (*Himantopus himantopus*) and Pied Avocet (*Recurvirostra avosetta*). Of all the 12 species recorded, Lesser Sand Plovers were not dominant in the region. Thane Creek is a productive habitat for shorebirds, but it is severely contaminated with garbage from both domestic and industrial sources [36].

The coastal zone of Sindhudurg district, Maharashtra, is of international importance for shorebirds and 31 species of migratory shorebirds and five species of resident shorebirds were recorded from there [19]. One Endangered species, the Great Knot (*C. tenuirostris*) and five near-threatened species, the Eurasian Oystercatcher (*Haematopus ostralegus*), Eurasian Curlew, Curlew Sandpiper, Black-tailed Godwit and Bar-tailed Godwit (*L. lapponica*) were recorded. Lesser Sand Plover, Common Sandpiper and Kentish Plover accounted for 72% of Sindhudurg's shorebird population. Besides them, the other most common species were the Common Redshank, Common Greenshank and Greater Sand Plover (*C. leschenaultia*). The Broad-Billed Sandpiper (*Limicola falcinellus*) and Eurasian Oystercatcher were found only in beach habitats, whereas the Great Knot was observed only on mudflats [19].

In a report that check-listed 35 species of migratory shorebirds along the various habitats of Goa, the Great Knot has been enlisted as endangered and eight species have been enlisted as near threatened under the IUCN Red List, which include the Great Thick-knee (*Esacus recurvirostris*), Eurasian Oystercatcher, Eurasian Curlew, Bar-tailed Godwit, Black-tailed Godwit, Curlew Sandpiper, Buff-breasted Sandpiper (*C. subruficollis*) and Asian Dowitcher (*Limnodromus semipalmatus*) [37].

Kadalundi-Vallikkunnu Community Reserve (KVCR) is a coastal wetland comprising patches of mudflats, mangroves and sand beaches located in south western India from where diverse avian fauna has been reported [14] and a small population of Lesser Sand Plover, Whimbrel (*N. phaeopus*) and Common Redshank were recorded to be over-summering in the habitat [33]. The most abundant species recorded in KVCR is the Lesser Sand Plover, which constitutes about 4% of the global population [38,39]. Endangered shorebirds like the Great Knot; vulnerable species like the Eurasian Oystercatcher; and near-threatened species like Curlew Sandpiper, Bar-tailedGodwit, Black-tailedGodwit and Eurasian Curlew have been recorded from this important site [38]. However, an increase in relative humidity, air and water temperature and salinity over years resulted in a decrease in prey abundance, which in turn adversely affected the abundance of shorebirds and caused their departure delay [14].

In a study conducted in the Kole wetlands of Thrissur (Kanjany, Parappur, Chettupuzha and Enamavu), Kerala, from November 1998 to April 2001, 34 wader species were recorded out of which 20 belonged to shorebird species. A high abundance of Wood Sandpiper, Little Stint, Little Ringed Plover, Curlew Sandpiper, Common Sandpiper and Eurasian Curlew were observed at Kanjany. Among these, Wood Sandpipers were dominant, followed by Little Stint [18]. The highest number of Curlew Sandpipers was observed during December and January whereas a higher number of Eurasian Curlews was observed from November to December. Common Sandpipers were very low in the first and third years but in the second year, a high number of Common Sandpipers were observed, especially during the month of December. The highest number of Little Ringed Plovers and Little Stints were recorded in the month of November and Wood Sandpipers were

observed during November and December [18]. This can be attributed to the fact that the abundance of shorebirds is at its peak during December and January, that is, post the south west monsoon periods on the west coast of India. Soon after the south west monsoon, the area becomes ideal foraging and roosting grounds for these migrant shorebirds as the prey availability and accessibility will be higher and the availability of exposed mudflats during low tides also supports them.

Out of a total of 39 species of shorebirds, 34 migrant species were reported from Changaram wetland in Kerala from 2018 to 2019 [34]. Most of the species were observed during the returning period of migration and they included IUCN enlisted endangered, Great Knot and near-threatened shorebird species such as the Black-tailed Godwit, Curlew Sandpiper, Eurasian Curlew. As per the study, 14 species of shorebirds were categorized as dominant and regular winter visitors, three were classified as unusual and the remaining 22 as rare. The Black-tailed Godwit, the most dominant species, attained its peak count in April, and the Wood Sandpiper, the second most dominant species, reached its peak count in May [34]. At Changaram wetlands, an agriculture system called Pokaalli is practiced, where paddy and shrimp/fish culture is practiced alternatively in the same wetlands, that is, paddy culture from May to October and shrimp/fish culture from November to April. At the end of April, after fish culture, the water is drained out completely from the wetlands, which leads to the exposure of mudflats rich in invertebrate fauna which in turn attracts the migratory shorebirds like the Black-tailed Godwit and Wood Sandpiper and serves as foraging grounds for them [34].

Each wintering ground studied reveals its own characteristic patterns in terms of species diversity and abundance of shorebirds. It could be an outcome of the macro and micro level habitat characteristics that need to be further investigated. To treat these wintering grounds with unerring conservation strategies such studies need to be conducted in depth. The species composition of shorebirds was found to vary across sites and months and is largely influenced by the heterogeneity of estuarine habitat, seasonality, and anthropogenic pressure [14,18,19,34]. In Sindhudurg district, Maharashtra, the winter season was characterized by significant richness and abundance of shorebirds, especially in the mudflats (73% of the birds counted in Sindhudurg district were encountered on mudflats) and the abundance values declined slowly from February to May [19].

Shorebirds such as Plovers, Sandpipers, Stints, Greenshanks and Redshanks were observed arriving along the Sindhudurg coast from August and continued until October. Populations of wintering shorebirds such as Plovers, Sandpipers, Greenshanks and Redshanks reached their peak during the November to February period. Consequently, almost all shorebirds, except a few over-summering individuals (Lesser Sand Plover, Greater Sand Plover, Common Redshank, Common Greenshank, Whimbrel and Common Sandpiper) departed to the breeding grounds toward the end of May.

In KVCR, a significant relationship between the number of polychaetes and feeding Lesser Sand Plovers was documented at mudflats with their population at its peak during December [40]. Even though shorebirds prefer mudflats to mangroves and sandy beaches, a declining trend in the number of birds using mudflats, mangroves and shallow-water habitats at KVCR was reported whereas the number of shorebirds using sandy areas increased significantly (Figure 2) [38,41].

Long-term dynamic population studies on shorebirds have not been carried out anywhere on the west coast except at KVCR. A significant decline in the shorebird abundance in the KVCR wetland was reported from 2005 to 2019 [14]. Further, the overall count of shorebirds has declined catastrophically over the last 10 years (Figure 3) [14]. The abundance of Pacific Golden Plovers, Lesser Sand Plovers, Greater Sand Plovers, Kentish Plovers, Little Stints and Terek Sandpipers (*Xenus cinereus*) was observed to be declining significantly over the years. All these decline in numbers and shifts in habitat use were attributed to high predator pressure. Hence it is evident that apart from abiotic factors, biotic factors also play a major role in determining shorebird abundance, species diversity and distribution.

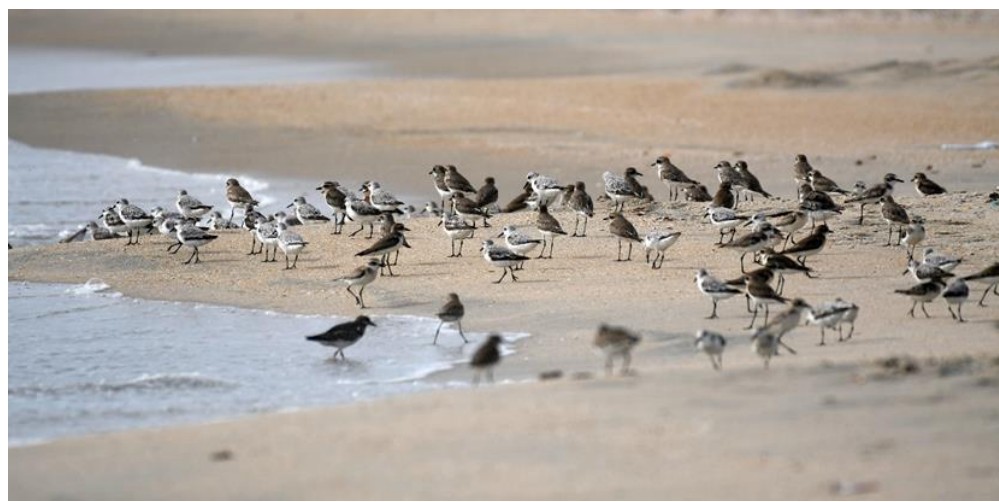

**Figure 2.** Lesser Sand Plover (*C. mongolus*), Sanderling (*C. alba*) and Ruddy Turnstone (*A. interpres*) at Puthiyappa beach in the west coast of India—Photo courtesy: C.T. Shifa.

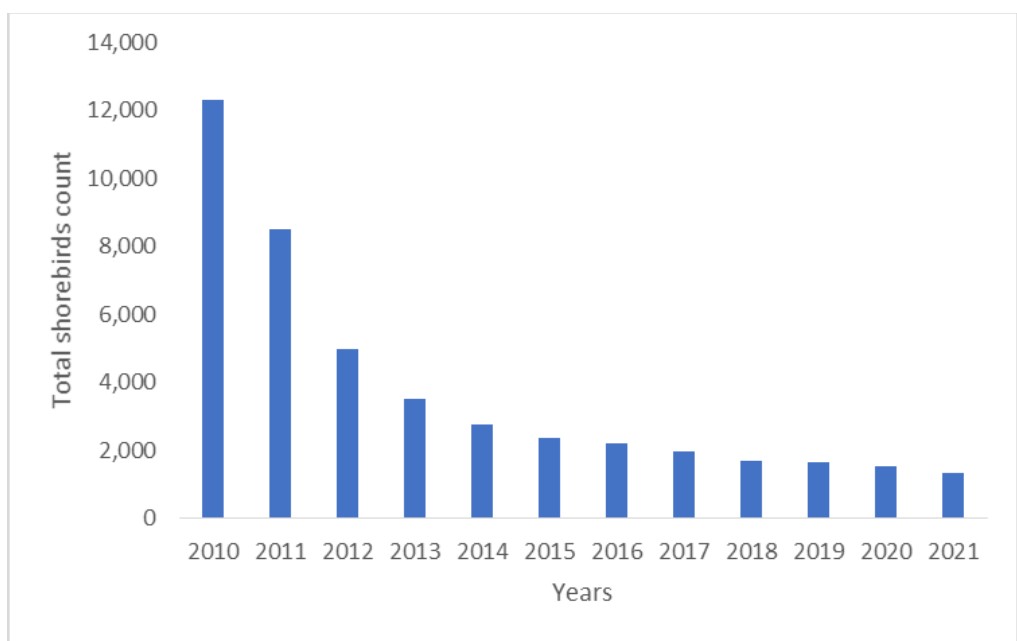

**Figure 3.** Variation in the counts of shorebirds over years at Kadalundi-Vallikkunnu Community Reserve, west coast of India—K.M. Aarif, (2010–2019) [14] and unpublished data 2020–2021.

The west coast of India is more productive than the east coast [42,43]. However, the studies of shorebirds on the West Coast are far fewer than those on the East Coast [14,19,34,38]. Therefore, there is an urgent need to develop robust conservation plans for shorebirds and to encourage in-depth studies on them along the West Coast.

*Distribution Pattern of Shorebirds on the West Coast*

The west coast of India extends from the Gulf of Cambay (Gulf of Khambhat) in the north to Cape Comorin (Kanyakumari) in the south, which touches six states such as Gujarat, Maharashtra, Goa, Karnataka, Kerala and Tamil Nadu. The west coast of India is copious with a variety of habitats such as sandy beaches, mudflats, mangroves, salt pan and tidal flats, which are preferred by shorebirds [44]. However, the studies on the distribution pattern of migrating shorebirds along the west coast are minimal. The records avail-

able regarding their distribution along the west coast are from Sindhudurg, Thane creek, Kadalundi-Vallikkunnu Community Reserve and Kole wetlands as mentioned earlier.

Gujarat, which has the longest coastline on India's west coast (about 1650 km), serves as an important stop-over and wintering grounds for many resident and migrant shorebirds [44–46]. Narara Island and Rozybundar coast in the Gulf of Kachchh have been recorded as important foraging and roosting sites for shorebirds, especially for Crab Plover, Eurasian Curlew, Black-tailed Godwit and Pied Avocet [47]. The Rozybundar coast was recorded with the lowest concentration of these species, which is believed to be due to higher anthropogenic pressure. Although the number of Eurasian Oystercatchers recorded in India has varied considerably over years, a constant and substantial proportion of the species has been observed from Byet Dwarka Island [35]. Shorebirds have also been observed from Jamnagar, which is further to the east of Byet Dwarka Island, however, the number of species found in Jamnagar was lower than that of Byet Dwarka Island [48].

The Maharashtrian Coast, which includes Mahul-Sewri Creek, Thane Creek wetlands and Navi Mumbai, is a well-known and significant wintering ground for migratory shorebirds because of its wide mudflats and shallow tidal beaches [45]. A small patch of mangrove ecosystem and mudflat of Thane has been found nurturing a number of shorebird populations. Further, the dominance of shorebirds along the rocky and sand beach shores of Akshi and Revdandanda (Maharashtra Coast) has also been reported [49]. These coasts shelter a large variety of gastropods and bivalves, which form the major prey for shorebirds and also have a uniform, gradual slope of the continental shelf, which enables smooth landing and provides favorable foraging grounds for shorebirds. The beaches, sandy mudflats, and mangroves of the Sindhudurg coast in Maharashtra have been identified as crucial wintering and stopover sites for many migrant shorebirds. Of the shorebird species observed there, 68% used the coast as their wintering site and 32% used the coast as a migratory stop-over site [19].

The coastal belt of Goa, which extends about 133 km, with its sandy bays, beaches, rocky headlands, saltpans and saline and freshwater marshes, coastal mangroves, creeks and estuaries [50] provide foraging ground for many migratory shorebirds including some endangered and near threatened species under IUCN Red List.

The Kole wetland, which is located below the sea level, is known for paddy cultivation and serves as important wintering and stop-over grounds for many transcontinental migrant shorebirds, in Kerala [18]. Along the entire west coast, KVCR in Kerala has been extensively studied for population and abundance of shorebirds and it is the most significant wintering and stop-over site for many migrant shorebirds where the shorebirds reach their maximum count at low rainfall periods [13,38].

Changaram wetlands, one of the least explored Pokaalli agro-ecosystem in central Kerala, is attracting migratory shorebirds in recent years [34]. From there, 33 species have been identified as winter visitors, two species, the Kentish Plover and Great Thick-knee, were recorded as local migrants and four species were listed as residents but non-breeding individuals. As migratory shorebirds use such fragile habitats as their crucial wintering and stop-over grounds, these habitats also need to be protected.

The distribution and abundance of shorebirds are highly supported on the west coast by various factors when compared to the east coast. Because of the high levels of sand and sandy silts, the sediments on the west coast have a loose texture, which is favorable for the dwelling of macrobenthic polychaetes [43]. These polychaetes, being the key prey items for the foraging shorebirds, attract more diverse species and numbers of migratory or locally moving and resident species of shorebirds [40,51–54].

## 3. Species Diversity, Abundance and Population Trends of Shorebirds on the East Coast

The eastern coast of India stretches northward along the Coromandel Coast to West Bengal. Its width ranges from 80 to 100 km, which is more extensive and broader than its western counterpart. It is an aggradational plain and is characterized by sea beaches,

lagoons and offshore bars. Dominant shorebird groups recorded from the west coast are Plovers followed by the Common Greenshank and Common Redshank [13] while those of the east coast are Little Stint, Black-tailed Godwit, Bar-tailed Godwit and Curlew Sandpiper (Figure 4) [16,21]. This is due to the narrow intertidal mudflats dominated by sand at the west and muddy substrate at the east [55].

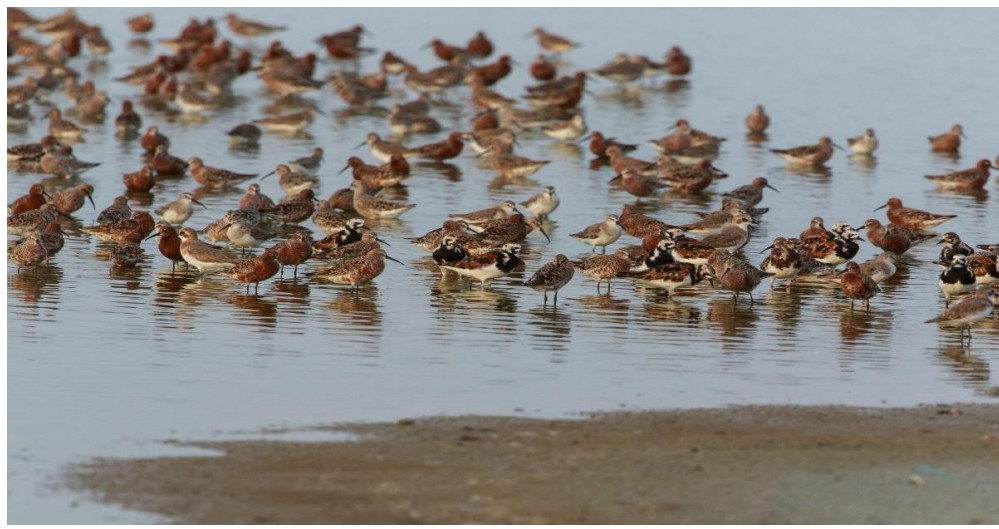

**Figure 4.** Mixed flock of Curlew Sandpiper (*C. ferruginea*), Lesser Sand Plover (*C. mongolus*), Greater Sand Plover (*C. leschenaultia*) and Ruddy Turnstone (*A. interpres*) observed at Gulf of Mannar at the east coast of India—Photo courtesy H. Byju.

The diversity and abundance of shorebirds in the Sundarbans, a protected biosphere, is marked by the records of critically endangered Spoon-Billed Sandpipers [56–61] and a total of 4000 small shorebirds [57], the most abundant species being Lesser Sand Plover, followed by Kentish Plover, Common Redshank and Greater Sand Plover. Three notable species recorded include the Eurasian Curlew, the Great Thick-Knee (both are near-threatened) and the Eurasian Oystercatcher [20]. Further, 37 species of shorebirds were listed from West Bengal from 2018 December to August 2020 [62].

A total of 800,000 birds were reported from Chilika lake during 2002–03, the Black-tailed Godwit (48,000 numbers) being the predominant species [63]. Large breeding colonies of Black-Winged Stilt along with Oriental Pratincole (*Glareola maldivarum*) and Kentish Plover occur in much greater numbers. The Presence of Spoon-Billed Sandpipers and Asian Dowitchers, though in small numbers (10–15) [64], in the checklist of birds enhances the importance of Chilika Lake. During a winter survey, nearly 3000 Black-tailed Godwits were recorded [65] from Bhitarkanika, Odisha.

Out of 49 species of migrant shorebirds in India, 34 were reported at Pulicat Lake [16]. A total of 88 species of water birds have been recorded from Pulicat Bird Sanctuary [66], of which shore birds represented a major portion.

In Vedaranyam, a swamp of the Point Calimere, 47 species of shorebirds have been recorded, the commonest being Lesser Sand Plover, Marsh Sandpiper, Little Stint and Curlew Sandpiper. The population of Little Stint during the peak season reaches 25,000 and that of each of the other three species is approximately 15,000 [22]. Up to four Spoon-Billed Sandpipers were recorded annually until 2004 [67] and the endangered Nordmann's Greenshank (*T. guttifer*) has also been recorded at Point Calimere [45].

A study reported 27 species of shorebirds at Pichavaram in 2021 [68], in which the Little Stint showed the highest density and the Bar-tailed Godwit showed the lowest density. Gulf of Mannar (GoM), a marine biosphere reserve, is known for the populations of shorebirds such as Eurasian Oystercatcher, Grey Plover, Lesser Sand Plover, Greater Sand Plover, Bar-tailed Godwit, Eurasian Curlew and Crab Plovers who prefer sand flats [69]. The avifaunal distribution of all 21 islands of GoM has been documented recently (Figure 5) [70]. The

population of most of the abundant species recorded during the 1990s had significantly reduced, which can be attributed to the degradation of habitat, especially in the Manoli islands [71,72]. Red Knot *C. canutus* (a regular winter visitor), Great Knot (rare bird), Crab Plovers and Bar-tailed Godwit (in hundreds) Sanderling (regular common winter migrant) are the main attractions of this site [69,73,74]. The Kentish Plover and Indian Thick-Knee (*Burhinus indicus*) use the Mandapam area as their breeding site [75]. The Gulf of Mannar lies within the passage of many migrants such as Black-tailed Godwit and Broad-Billed Sandpiper. Hence this site forms an extremely important link for migrant shorebirds along with Chilika Lake in Orissa and Point Calimere in Tamil Nadu on the east coast of India.

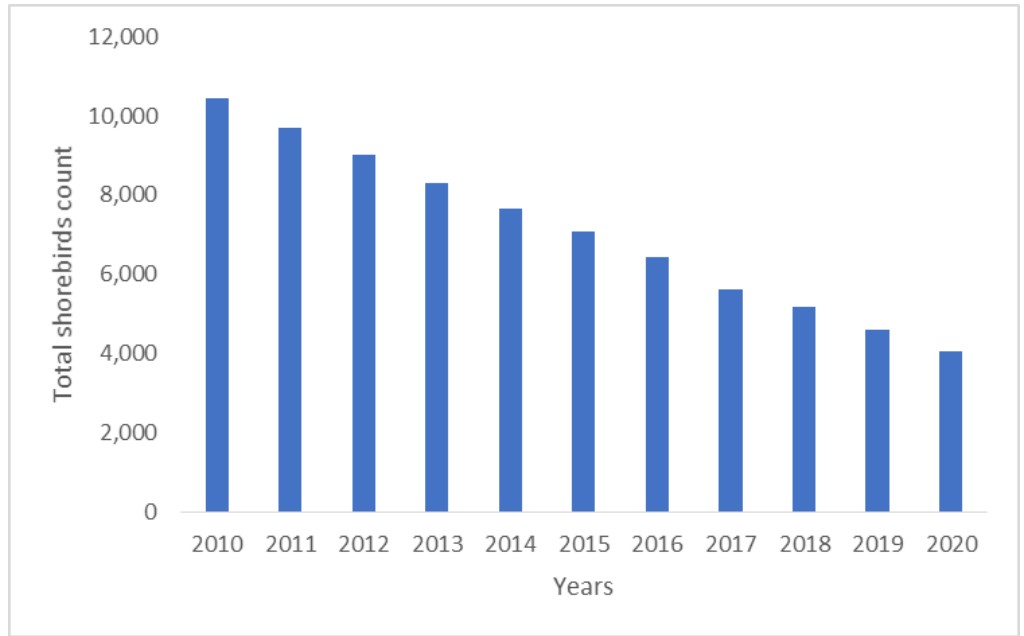

**Figure 5.** Variation in shorebirds' count over years at Gulf of Mannar, east coast of India—H. Byju, unpublished data.

Coastal zones are efficiently utilized by avian fauna for millennia and many shorebird species are now threatened. Therefore, systematically monitoring the dynamics of shorebirds and their habitats over long-term periods is important for improving management strategies for their effective conservation [76,77]. Understanding the long-term habitat changes through systematic demographic monitoring [78] that helps in the statistical analysis of shorebird populations underpins conservation management [79]. The Gulf of Mannar [69], Point Calimere [22], Pichavaram [21], Pulicat Lake [16], and Chilika Lake [80] are significant stopover and wintering grounds for migrating shorebirds on India's east coast. Out of the 49 species of migrating shorebirds recorded in India [81], 34 are found on the east coast [16,82].

Lesser Sand Plover, Little Stint, Common Greenshank, Marsh Sandpiper, Common Redshank, Pacific Golden Plover and Eurasian Curlew are among the first to arrive (sometimes as early as July) while, Wood Sandpipers and Common Sandpipers can be seen by the end of August and Grey Plover, Little Ringed Plover, Spotted Redshank (*T. erythropus*), Curlew Sandpiper, Ruff and other species can be found in September, and Black-tailed Godwit and Pied Avocet can be found in October. The mudflats host the highest population of shorebirds in Chilika Lake because Nalabana Island is exposed there from December to May during the dry season. Small and medium-sized shorebirds, which are rare until December, reach their peak in late January. From November to March, big and long-legged Black-tailed Godwits have been observed in significant numbers [64] mostly using the Mangalajodi area as a foraging ground [83].

The Lesser Sand Plovers reach their highest populations at Nalabana Island on Chilika Lake in January and February. Ruffs, Little Stints, and Curlew Sandpipers are common shorebirds in this area. Throughout the year, very few individual Whimbrels were observed. Rare winter migrants include the Bar-tailed Godwit and Green Sandpiper (*T. ochropus*), the latter of which is typically seen in small puddles around Chilika Lake's Parikud, Mangalajodi and Gurubai regions. The Broad-billed Sandpiper is an occasional winter visitor and the uncommon winter visitors include the Terek Sandpiper, Temminck's Stint, Asian Dowitcher and Ruddy Turnstone. The Greater Sand Plover is an uncommon and irregular migrant, whereas Dunlin (*C. alpina*) and Pied Avocets are regular winter visitors [83].

Many shorebirds that were previously recorded from Chilika are no longer there. The last authenticated record of the critically endangered Spoon-billed Sandpiper was a bird that was ringed in 1981. Eurasian Oystercatcher and Common Ringed Plover (*C. hiaticula*) were not present after 2000, and there were no recent records of Sanderling [83] but they were sighted in 2012.

The migration season typically begins at Pulicat Lake on the east coast in the first week of August, and it lasts until the second week of April to the first week of May. Lesser Sand Plover, Black-tailed Godwit, Whimbrel, Common Redshank, Common Greenshank, Wood Sandpiper, Common Sandpiper, Little Stint and Ruff are the first species to arrive in August while Grey Plovers, Curlew Sandpiper and Red-necked Phalarope (*Phalaropus lobatus*) arrive in September and Marsh Sandpiper and Temminck's Stint arrive by the northeast monsoon (October). It is in January when shorebird populations are at their highest [16]. Lesser Sand Plover, Black-tailed Godwit, Little Stint, Curlew Sandpiper and Ruff are the most prevalent shorebirds in this area [64].

With a few records at Point Calimere [84], Pulicat [85] and Kolkata [86], the Great Knot was previously considered a "rare winter visitor" to the east coast of India [86]. The migratory and sighting records at Chilika shows that it is an uncommon and regular visitor here. Similarly, Long-toed Stint (*Calidris subminuta*) used to be regular and uncommon [86], recorded from Assam, Bihar and Andaman Islands. However, another study [75] revised the status to "rare winter visitor" as one bird was ringed at the Gulf of Mannar and 13 birds at Point Calimere. Another species to be mentioned is the Red Knot, a regular winter visitor to Point Calimere and the Gulf of Mannar [87,88]. The recent ringing records in Chilika further confirm the extension of its wintering range along the east coast [83]. Recent sightings were also reported from the Pallikaranai wetlands of Chennai and Kanyakumari saltpans [83].

Among the shorebirds wintering at the Gulf of Mannar, Lesser Sand Plover, Curlew Sandpiper and Little Stint were common in all the six habitats studied during 1985–1988 and their numbers exceeded over 1000 in all the selected study sites. The dominance of these three species was seen in all of the coastal wetlands studied by the BNHS, including the key east coast sites of Point Calimere, Kaliveli, Pulicat and Chilika [87]. The reduction of these three species in all of those wetlands is similar to all wetlands along the east coast and is in line with the global trend of decline in the shorebird population. However, the proportion of the decline in the Gulf of Mannar is significant when compared to other locations (Figure 5) [72].

*Distribution Pattern of Shorebirds on the East Coast*

Large stretches of intertidal regions, mangroves, sea grasses, coral islands, coastal swamps, mudflats, lagoons, and other similar ecosystems can be found throughout the east coast. However, there are some amazing shorebird populations along the beaches, as seen at Chilika Lake and Bhitarkanika in Orissa, Pulicat Lake in Tamil Nadu and Andhra Pradesh, Point Calimere and Gulf of Mannar in Tamil Nadu, and Sundarbans (IBA) in West Bengal. Shorebirds were also observed from different regions of the east coast such as Thanjavur, Chennai, Ramanathapuram, Nagapattinam, Pondicherry, Karaikal and Kolleru Lake [89]. The studies of bird ringing by BNHS at Point Calimere had thrown light on the migratory paths of many shorebird species to and from the east coast. The records available regarding

their distribution along the east coast are from Sundarbans, Chilika Lake, Bhitarkanika, Point Calimere, Pichavaram, Kaliveli and the Gulf of Mannar as mentioned earlier.

The Sundarbans, a protected biosphere, is the largest single tract of tidal mangrove forests and vast saline mudflats in the world, covering over 9360 sq. km (the Indian area 4264 sq. km). Besides having records of many wintering shorebirds, Sundarbans is an important breeding site for the Great thick-Knee [90]. It is a key wintering site for shorebirds of which at least nine species occur in numbers exceeding the internationally recognized 1% threshold for site importance. Extrapolation to all areas of suitable habitat suggests that the total population was 40,000 shorebirds [57]. Almost 80% of the recorded species were Arctic breeding long-distance migrants such as Little Stint and Curlew Sandpiper. Recently an Oriental Plover (*C. veredus*) was also recorded here [62].

The largest brackish water wetland in India, the Chilika Lake, is an important wintering ground for shorebirds on the east coast. The various habitats in the Lake include marshes, mudflats, freshwater pools and areas of open water with varying depths and salinity. It is the largest wintering ground for migratory waterfowl in India [83]. The lake has been included in Ramsar sites. The Nalabana Island on Chilika Lake primarily serves as a staging area for the smaller shorebirds during the springtime northward migration [64] as it gets exposed only in December. For the small shorebirds, the lake serves mainly as a staging area during the northward migration in spring. Black-tailed Godwits are present in good numbers from November to March. The Asian Dowitcher is also recorded in a few numbers [64].

Bhitarkanika, Odisha, situated in the delta formed by Brahmani and Baitarani rivers, has one of the finest patches of mangroves and a coastline of 35 sq. km. The vast stretches of intertidal zones along the Gahirmatha coast attract shorebirds and are known for a large population of Black-tailed Godwits [65].

Pulicat Lake on the Andhra Pradesh-Tamil Nadu border is an extensive brackish to saline lagoon with associated marshes covering an area of 720 sq. km [91]. Most of the shorebirds are distributed over the extensive mudflats along the Sriharikota-Sullurpet Road near Tada in the southwestern part of the lagoon. It is an important wetland for migratory shorebirds and is identified as a coastal flyway used by a number of pelagic and coastal migrants linking Point Calimere, Tamil Nadu and Chilika, Odisha.

Point Calimere (Vedaranyam Swamp) at the Bay of Bengal seaboard of Thanjavur is another important wintering quarters for shorebirds, with the great Vedaranyam swamp of Point Calimere, stretching for about 48 km from east to west, parallel to the Palk strait that connects India and Sri Lanka and separated by a sandbank, harbors a large number of shorebirds [92]. The commonest shorebirds are Lesser Sand Plover, Marsh Sandpiper, Little Stint and Curlew Sandpiper. The saltpans near the Sanctuary and Siruthalaikadu are areas where we find a good congregation of migrating shorebirds.

Another vital stopover site on the east coast of Southern India is Pichavaram Mangrove Forest. It is situated along the CAF routes of migratory shorebirds and it provides sufficient nutrients for the shorebirds [22,68].

The Kaliveli estuary, an Important Bird Area, is among the largest and one of the most important brackish water wetlands in southeastern India after Point Calimere, in Tamil Nadu. It consists of mudflats, marshes, reed beds, saltpans and open shore, attracting a variety of bird species including shorebirds. Kalveli has significant geographic populations of Little Stint [93,94].

The Gulf of Mannar, due to its proximity to Sri Lanka, is considered an important part of CAF for migratory birds. The Gulf of Mannar Biosphere Reserve comprises an island ecosystem of 21 islands extending from Rameswaram Island in the North to Tuticorin in the South. The major shorebird congregations are found in Dhanushkodi lagoon on Rameswaram Island and Pillaimadam lagoon on the mainland near Mandapam [69]. Because of the Extensive areas exposed during low tide, the Gulf of Mannar is significantly important on the east coast for the sandflat preferring shorebirds such as Great Knot, Red Knot, Bar-tailed Godwit, Terek Sandpiper, Greater Sand Plover, Sanderling, Oystercatcher,

Crab Plover, Eastern Curlew (*Numenius madagascariensis*), Whimbrel and Ruddy Turnstone, which were relatively less common in other three major sites like Chilika, Pulicat and Point Calimere [75]. Recent observations in GoM [70,72] found that the distribution of shorebirds is restricted only to the Dhanushkodi lagoon due to anthropogenic activities for economic development and human settlements. Few of the previous bird congregation areas like Pillaimadam and Chinna Palam had been partially or fully converted for fisheries and hence a decline in bird counts in these areas is reported. However, few saltpans of GoM are supporting the over-summering of some of the migratory shorebirds in this important site as well as new conservation steps on declaring bird sanctuaries are also under consideration by the authorities.

A triangular area covering 10 to 20 km, which is situated between Point Calimere, Gulf of Mannar and the adjacent Sri Lankan areas, is uninhabited and undisturbed. The species composition there varies greatly from other sites on the east coast and our observations in the past decade show that Crab Plovers are restricted to the Manoli and Hare Islands of the Gulf of Mannar on the east coast due to the increased availability of fiddler crabs in the islands, and they move to the Mannar regions of Sri Lanka. Similarly, it has been found that many shorebird species were found varying across sites and months, and are largely influenced by the rainfall pattern and climatic changes with unexpected cyclones and storms hitting the coast. In addition to these ecological and environmental factors, anthropogenic interventions of developmental activities and ecotourism-related disturbances affecting the feeding grounds of shorebirds have resulted in their population decline.

## 4. Over-Summering Shorebirds on the East and West Coast of India

Limited information is available regarding Indian coasts on over-summering shorebirds. However, seven over-summering species were documented from KVCR over a 14-year study from 2005 to 2018, which include Lesser Sand Plover, Whimbrel, Greater Sand Plover, Kentish Plover, Common Sandpiper, Ruddy Turnstone and Pacific Golden-Plover [17]. Some of these species have also been over-summering in the coastal zone of Sindhudurg district, Maharashtra such as Lesser Sand Plover, Greater Sand Plover, Common Redshank, Common Greenshank, Whimbrel and Common Sandpiper [19]. Species such as the Black-tailed Godwit, Common Sandpiper, Wood Sandpiper, Marsh Sandpiper and Pacific Golden Plover were reported to be over-summering in Changaram wetland, Kerala during June and July [34]. Mudflats were found with the highest mean abundance, species richness, species diversity and evenness of over-summering species when compared to mangroves and sand beaches [17].

Grey Plover, Lesser Sand Plover, Whimbrel, Eurasian Curlew, Common Redshank, Common Greenshank, Terek Sandpiper, Curlew Sandpiper and Broad-Billed Sandpiper are a few species that regularly over-summer in Point Calimere, east coast [95]. The over-summering of Ruff, Pied Avocet and Common Sandpiper is in the first summer records from South India. Similarly, the record of Red Knot at Point Calimere is the first summering record from India [95]. Further, Grey Plover, Lesser Sand Plover, Whimbrel and Eurasian Curlew are a few species that are over-summering in Chilika Lake [83]. Over-summering species registered from Pulicat Lake are the Pacific Golden Plover, Eurasian Curlew, Black-tailed Godwit, Marsh Sandpiper and Common Sandpiper [16]. As the intertidal zones of the islands of the GoM offer ideal foraging sites throughout the year for migratory shorebirds, a small population of 12 species of shorebirds spends the summer as well [75].

## 5. Conservation Importance of Shorebirds in the Indian Coasts

Since migratory shorebirds connect different countries around the globe, they are considered the ideal ecological indicators at global, regional and local scales [39,96]. In India, though they use diverse habitats, migratory shorebirds prefer coastal zones as their wintering grounds as the population abundance of their key prey species is higher in marine habitats than freshwater habitats and also the freshwater bodies may dry up during spring. The increasing decline in the diversity and abundance of shorebirds can be attributed

to innumerable disturbances such as habitat deterioration, prey depletion and predation pressure across the flyways, particularly at the wintering and stop-over sites [19,39,96,97].

Shorebirds wintering on the west coast of India encounter serious environmental issues, particularly caused by anthropogenic activities such as the dumping of organic and plastic wastes, indiscriminate application of weedicides and pesticides, and extensive sand mining [19,33,34]. The anthropogenic debris like glass, plastics, discarded fishing gear nets, and metal rings cause injuries to shorebirds, restricting their way back to breeding grounds and are consequently doomed to death either by predation or by decreased intake of food upon the injury [16,98,99]. The removal of nutrient-rich topsoil during sand mining which contains polychaetes, the major prey of shorebirds, can cause them to disappear from the area and further, the declining abundance of phytoplankton and zooplankton, which has far-reaching effects on higher trophic levels, particularly shorebirds [15].

KVCR, though an important stop-over and wintering ground for shorebirds along the west coast, witnessed a drastic decline in the diversity of shorebirds over years, which can be attributed to crucial factors like incursion of mangroves, changes in sediment quality, expansion of sand bed, shrinking of mudflat and decreasing mudflat thickness [39]. Moreover, the dumping of organic waste materials attracts House crows, Brahminy Kites and Jackals, which are the major predators of shorebirds, enhancing the risk of predation on them [33,100].

Thane Creek is yet another productive wintering habitat for shorebirds along the west coast which is severely contaminated with both household and industrial garbage and seeks immediate conservation actions to support thousands of birds including migratory shorebirds who still depend on this habitat [36].

The unchecked spreading of grass and other weeds like Salicornia into the mudflats along with unscientific dredging of Nalabana Island of Chilika is a major cause of habitat degradation and it should be addressed properly and with utmost importance. The incursion of mangroves and the new vegetative structures in and around the Manoli islands of the Gulf of Mannar reduced the area of mudflats considerably over three decades [72]. Developmental activities and ecotourism at Dhanushkodi lagoon are also having a negative impact on the population of shorebirds in GoM. The continuous extreme siltation which converts shallow areas into sandflats, the 630 MW North Chennai thermal power station (NCTPS), Ennore satellite Port Project and Petrochemical Park are serious threats to the Pulicat Lake ecosystem and to the shorebirds as well [16].

There are several shorebird species that are common to both coasts while some others are confined to a particular coast. Many of these are categorized by the IUCN as endangered species. According to our case studies at two potential wintering sites—KVCR [14,99] (Figure 3) and Gulf of Mannar [72] (Figure 5) from the west and east coast, respectively, an overall decline in the population of shorebirds has been observed. However, KVCR is facing a catastrophic steep declining trend in population over a decade (Figure 3) whereas the declining rate of shorebirds at the Gulf of Mannar is steady (Figure 5), which points out how seriously the shorebirds are affected by various environmental issues along the west coast. Shorebirds are considered ecosystem indicators since they can react even to the narrow changes undergoing in the environment [101,102]. Systematic monitoring of the population status, spatio-temporal distribution patterns and habitat use of shorebirds can shed light on the altering environmental parameters and their impacts on the ecosystem, which will be instrumental in implementing issue-specific management plans, particularly on the west coast where the shorebirds are severely affected and are least explored compared to the east coast. Hence, the west coast demands more and extensive scientific investigation as well as conservation strategies to protect the shorebird community and their habitats [19,33,103].

The shallow waters of the west coast facilitate better penetration of sunlight helping the plankton community to flourish, which is reflected in higher trophic levels as well. The intense southwest monsoon also plays a key role in increasing the productivity of the west coast as evidenced by the diversity and abundance of the shorebird species. Nevertheless, in the case of the east coast, deep waters hinder sunlight penetration, which adversely affects

the plankton community, bringing down the population on all trophic levels. Furthermore, frequent natural calamities add to the habitat degradation on the east coast and this should be reflected in the diversity and abundance of the shorebird species being the top predators (Figure 6). However, the east coast appears to have a higher diversity and abundance of shorebird species than the west coast [14,16,19,81,82]. Thus, extensive systematic studies should be carried out on the west coast to explore its productivity.

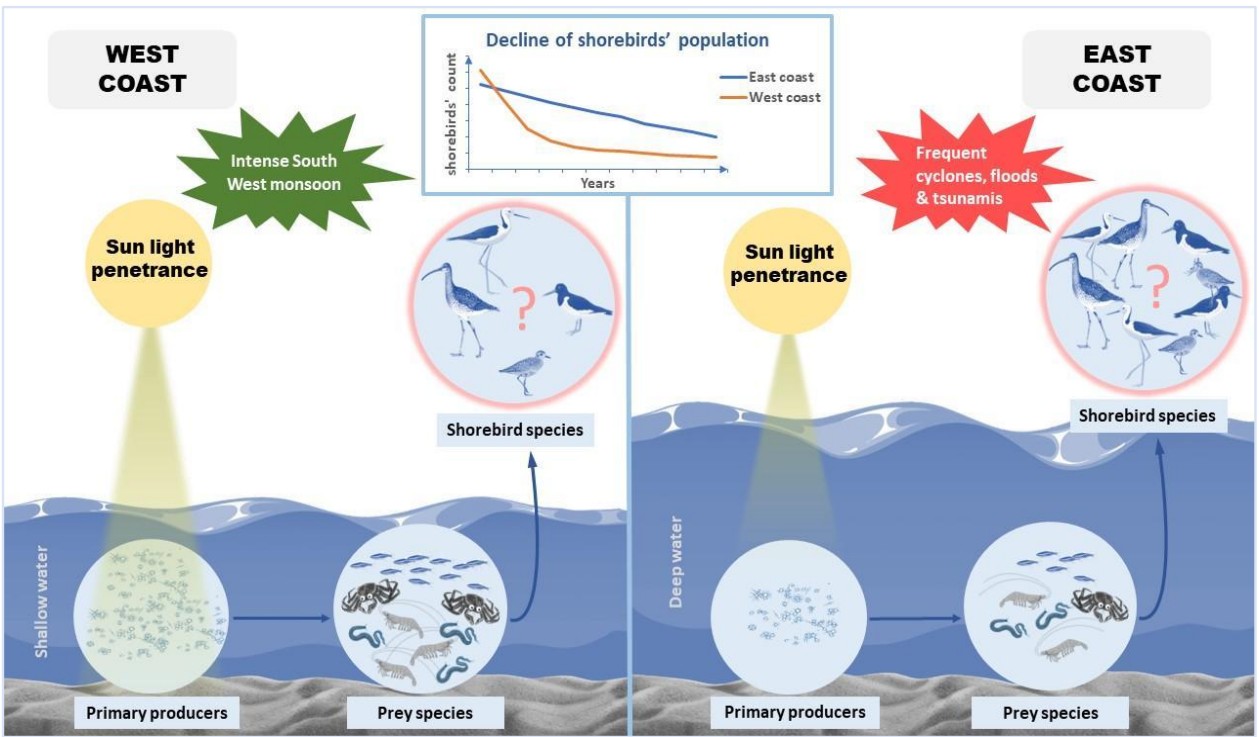

**Figure 6.** Ecosystem components, environmental variables and population trends in shorebirds species in the west and east coast of India.

## 6. Conclusions

Of the 215 shorebird species (belonging to 14 families) identified in the world, the east and the west coast of India support 48 and 41 species, respectively (Tables 1 and 2). The varying diversity and abundance of shorebirds along the Indian coast might be due to the distinctive topographical as well as biotic and abiotic factors characterizing the east and west coast of India.

The west coast of India claims a greater proportion (71.1%) of the continental shelf but a lesser proportion of the coastline (42.8% in length) compared to the east coast. Along with the shallow continental shelf, the southwest monsoon plays a critical role in altering the local environmental features of the west coast by influencing its temperature, salinity, dissolved oxygen and nutrients; and is more productive than the east coast in terms of primary, secondary and tertiary productivity. Consequently, the west coast is rich in biodiversity and abundance than the east coast and it is primarily reflected in the phytoplankton diversity especially in diatoms and dinoflagellates, as well as in fish populations and finally in top predators including shorebirds.

**Table 1.** List of shorebirds documented from east and west coast of India.

| S. No | Common Name | Scientific Name | IUCN Status [104] | Migration Status * | Kadalundi * | Kole Wetland [105] | Sindhudurg * | Chilika * | Pichavaraum * | Gulf of Mannar * | Point Calimere * | Pulicat Lake * |
|---|---|---|---|---|---|---|---|---|---|---|---|---|
| | | | | | | West Coast | | | | East Coast | | |
| 1 | Lesser Sand Plover | *Charadrius mongolus* | LC | WV | Common | Uncommon | Common | Common | Common | Common | Common | Common |
| 2 | Greater Sand Plover | *Charadrius leschenaultii* | LC | WV | Common | - | Common | Uncommon | - | Common | Common | - |
| 3 | Little Ringed Plover | *Charadrius dubius* | LC | R/LM | Uncommon | Common | Uncommon | Common | - | Common | Common | Common |
| 4 | Kentish Plover | *Charadrius alexandrinus* | LC | WV | Common | Uncommon | Common | Common | - | Common | Common | Common |
| 5 | Pacific Golden Plover | *Pluvialis fulva* | LC | WV | Common | Uncommon | Common | Common | - | Common | Common | Common |
| 6 | Grey Plover | *Pluvialis squatarola* | LC | WV | Uncommon | - | Common | Common | - | Common | Common | Uncommon |
| 7 | Red-wattled Lapwing | *Vanellus indicus* | LC | R | Common | Common | Common | Common | Common | Common | Common | Common |
| 8 | Yellow-wattled Lapwing | *Vanellus malabaricus* | LC | R | Common | Common | Common | Common | Common | Common | Common | Common |
| 9 | Common Snipe | *Gallinago gallinago* | LC | WV | Rare | Common | - | Common | - | - | - | Uncommon |
| 10 | Black-tailed Godwit | *Limosa limosa* | NT | WV | Uncommon | Uncommon | - | Common | Common | Uncommon | Common | Common |
| 11 | Bar-tailed Godwit | *Limosa lapponica* | NT | WV | Common | Uncommon | - | Uncommon | - | Common | Uncommon | - |
| 12 | Whimbrel | *Numenius phaeopus* | LC | WV | Common | Uncommon | Common | Rare | Uncommon | Common | Uncommon | Rare |
| 13 | Eurasian Curlew | *Numenius arquata* | NT | WV | Common | Common | Common | Common | - | Common | Uncommon | Common |
| 14 | Spotted Redshank | *Tringa erythropus* | LC | WV | Rare | - | - | Uncommon | - | Rare | Uncommon | Common |
| 15 | Common Redshank | *Tringa totanus* | LC | WV | Common | Common | Common | Common | Common | Common | Common | Common |
| 16 | Marsh Sandpiper | *Tringa stagnatilis* | LC | WV | Uncommon | Common | - | Common | Common | Common | Common | Common |
| 17 | Common Greenshank | *Tringa nebularia* | LC | WV | Common | Common | Common | Common | Common | Common | Common | Common |
| 18 | Green Sandpiper | *Tringa ochropus* | LC | WV | Uncommon | Common | Rare | Rare | - | Common | Uncommon | - |
| 19 | Wood Sandpiper | *Tringa glareola* | LC | WV | Uncommon | Common | Rare | Common | - | Common | Common | Common |
| 20 | Terek Sandpiper | *Xenus cinereus* | LC | WV | Common | Common | Uncommon | Uncommon | - | Common | Common | - |
| 21 | Common Sandpiper | *Actitis hypoleucos* | LC | WV | Common | Common | Common | Common | Common | Common | Common | Common |
| 22 | Ruddy Turnstone | *Arenaria interpres* | LC | WV | Common | - | Uncommon | Uncommon | Common | Common | Uncommon | Rare |
| 23 | Great Knot/ Eastern Knot | *Calidris tenuirostris* | EN | WV | Rare | Uncommon | Rare | Uncommon | - | Uncommon | Common | - |
| 24 | Sanderling | *Calidris alba* | LC | WV | Common | Uncommon | Uncommon | Rare | - | Rare | Common | - |

**Table 1.** *Cont.*

| S. No | Common Name | Scientific Name | IUCN Status [104] | Migration Status * | Kadalundi * | Kole Wetland [105] | Sindhudurg * | Chilika * | Pichavaraum * | Gulf of Mannar * | Point Calimere * | Pulicat Lake * |
|---|---|---|---|---|---|---|---|---|---|---|---|---|
| | | | | | | West Coast | | | | East Coast | | |
| 25 | Little Stint | *Calidris minuta* | LC | WV | Common | Common | Common | Common | Common | Common | Common | Common |
| 26 | Temminck's Stint | *Calidris temminckii* | LC | WV | Rare | Common | - | Uncommon | Uncommon | Rare | Uncommon | Common |
| 27 | Dunlin | *Calidris alpina* | LC | WV | Common | - | Uncommon | Common | - | Uncommon | Uncommon | - |
| 28 | Curlew Sandpiper | *Calidris ferruginea* | NT | WV | Common | Common | Uncommon | Uncommon | Common | Common | Common | Common |
| 29 | Broad-billed Sandpiper | *Limicola falcinellus* | LC | WV | Common | Common | Uncommon | Uncommon | - | Uncommon | Common | - |
| 30 | Ruff | *Philomachus pugnax* | LC | WV | - | - | - | Common | - | Rare | - | Common |
| 31 | Pied Avocet | *Recurvirostra avosetta* | LC | V | Rare | Common | - | Common | Common | Rare | Common | Common |
| 32 | Greater painted-snipe | *Rostratula benghalensis* | LC | R/LM | - | - | - | - | - | - | - | Rare |
| 33 | Eurasian Oystercatcher | *Haematopus ostralegus* | NT | V | Uncommon | - | Rare | - | - | - | Uncommon | - |
| 34 | Great Thick-knee | *Esacus recurvirostris* | NT | R/LM | Uncommon | - | Rare | Rare | - | Common | - | - |
| 35 | Crab Plover | *Dromas ardeola* | LC | V | Rare | - | Rare | - | - | Common | Rare | - |
| 36 | Swallow Plover/Small Pratincole | *Glareola lactea* | LC | R/LM | - | - | Common | Common | - | - | - | Rare |
| 37 | Oriental Pratincole | *Glareola maldivarum* | LC | LM/R | Rare | - | - | Common | - | - | | Rare |
| 38 | Black-winged Stilt | *Himantopus himantopus* | LC | LM | Rare | Common | - | Common | Common | Common | Common | Common |

LC—Least Concern, NT—Near Threatened, EN—Endangered; WV—Winter Visitor, LM/R—Locally migrant or Resident, R—Reside. * Migratory status and species status at study areas—based on field experiences as well as published information.

**Table 2.** Documentation of uncommon shorebird species from the west and east coast of India.

| S. No | Common Name | Scientific Name | IUCN Status [104] | Migration Status * | West Coast Status * | East Coast Status * |
|---|---|---|---|---|---|---|
| 1 | Buff-breasted Sandpiper | *Calidris subruficollis* | NT | WV | Rare | - |
| 2 | Asian Dowitcher | *Limnodromus semipalmatus* | NT | WV | Rare | Uncommon |
| 3 | Beach Thick-knee | *Esacus magnirostris* | NT | R/LM | - | Rare |
| 4 | Long-billed Plover | *Charadrius placidus* | LC | WV | - | Rare |
| 5 | Common Ringed Plover | *Charadrius hiaticula* | LC | WV | - | Rare |
| 6 | Pectoral Sandpiper | *Calidris melanotos* | LC | WV | - | Rare |
| 7 | Little Curlew | *Numenius minutus* | LC | WV | - | Rare |
| 8 | Red-necked Phalarope | *Phalaropus lobatus* | LC | WV | - | Rare |
| 9 | Nordmann's Greenshank | *Tringa guttifer* | EN | WV | - | Rare |
| 10 | Red Knot | *Calidris canutus* | NT | WV | Rare | Rare |
| 11 | Spoon-billed Sandpiper | *Calidris pygmaea* | CR | WV | - | Rare |

LC—Least Concern, NT—Near Threatened, EN—Endangered; CR—Critically endangered; WV—Winter Visitor, R/LM—Resident or Locally migrant. * Migratory status and coast wise species status—based on field experiences as well as published information.

Macrobenthic polychaetes, which are important prey for shorebirds, are most abundant along the nutrient-rich west coast, as a result of south-west monsoon wind-driven upwelling, the loose texture of sediment due to the high content of sand and the high-saline waters. Hence the high productivity of the west coast at every trophic level must be reciprocated by high species abundance and diversity of shorebird populations on the west coast than the east coast, which is frequently affected by various natural calamities such as cyclones, floods and tsunamis. However, from the available literature, it is deduced that the east coast has more diversity and abundance than the west coast (Table 1), which proves to be a paradox when the productivity of the west coast is taken into account. Hence there is a demand for more systematic and extensive, long-term investigations to unravel the spatial and temporal variations in the productivity parameters along the coastal zones of India from the shorebird diversity and abundance perspective.

Further research regarding anthropogenic interventions, which affect the ecosystem health that can eventually be a death knell to the shorebirds on the west coast, must be encouraged for a thorough understanding of the detrimental factors and for developing biodiversity conservation plans and waste management plans. Considering the east and west coast of India, implementing habitat-specific management plans at all the habitats along the coastal lines based on the observed pivotal threats to shorebird populations in the wintering sites may attract more diverse shorebird species, which in turn helps in biodiversity conservation.

**Author Contributions:** A.P.R.: writing, K.J.: writing, H.B.: writing, C.T.S.: writing, J.A.: writing, K.V.: writing, Y.X.: overall supervision, A.N.: overall supervision, S.B.M.: editing and supervision, K.M.A.: conceptualization, editing and overall supervision, K.A.R.: writing, editing. All authors have read and agreed to the published version of the manuscript.

**Funding:** This work is financially supported by Program for Advanced Research of UAE University (Grant no 31R463).

**Institutional Review Board Statement:** Not applicable.

**Informed Consent Statement:** Not applicable.**Data availability Statement:** Not applicable.

**Acknowledgments:** The authors acknowledge all the field assistants at Kadalundi-Vallikkunnu Community Reserve and Gulf of Mannar sites, helping to better understand how shorebird populations and habitats change over time.

**Conflicts of Interest:** The authors declare no conflict of interest.

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
