# Peer review of "The Paradox of Shorebird Diversity and Abundance in the West Coast and East Coast of India: A Comparative Analysis"

_diversity, doi:10.3390/d14100885_

Round 1

Reviewer 2 Report

Review of the paper „The paradox of shorebird diversity and abundance in the west coast and east coast of India: A comparative analysis”

The article is interesting and presents the current state of knowledge on bird migrants on the west and east coasts of India. There is no general justification for the value of these areas for migrants and the importance of wetland birds in other continents and other parts of the globe compared to migratory routes, although I appreciate the brief description of other areas where birds migrate so far. However, it would be valuable to link them better with India or even add why the article is about India, not India and other areas. There is no general information on why India is important due to global climate changes and what threats to migration concern these areas. Another point of mine is that the text is a bit chaotic in some places. The article would be better if it had a more reasonable structure, e.g. based on the ecological aspects of bird migration, than if in the current version it simply distinguishes between two opposite shores of the Indian peninsula. It is worth briefly justifying the choice of the current structure of the article if the authors decide to keep it. Below are some other suggestions on how to improve the article.

The question of using the generic names of birds once again after the first use of the full species name for the first time. Usually, full Latin genre names together with the genus appear in various places in the article, but I also found fragments where they appear only with the first letter of the genus name (i.e. the first segment of the genus name). It is worth paying attention to this.

The way to cite articles in the journal is such that it is better not to mention the names of the authors and who wrote what. It is better to simply provide substantive results without the author. You can get the impression that the reference number is like the year of publication in brackets in accordance with the citation standards in other publications.

I appreciate the work effort of the authors, but in some cases, the review resembles a set of small abstracts of various papers. I believe that a review should be a form of a serious popular science study, so it should have its original content, and not only summarize other works. I suggest you work on it.

Figures 3 and 5 – what is the source of this data?

There is no author or source under the photos. It is good practice to write full Latin generic names under the photographs alongside the English names.

Tables 1 and 2 – species status source is missing

Conclusions should be rewritten as excerpts from the review of other authors’ articles appear unnecessarily. Conclusions in a review paper should only be the authors’ thoughts on the described set of literature.

Reviewer 3 Report

The manuscript  reviewed previous studies about shorebirds wintering on the western coast and eastern coast of India and made a comparison of bird species richness and abundance between both coast, then the authors gave the suggestions of waterbird biodiversity conservation. The present study may benefit to bird conservation in India.

Here are some suggestions:

1 Please give some description of field survey methodology used for shorebirds count. 

2 Shordbird population decline is of worldwide concern. Please provide the causes and threats of shorebird decline in details if possible. Are these birds abundant and well protected in breeding area or stopover? 

3 Any food diversity and abundance or foraging habits of these waterbirds species?
